# The Complexity of Food Provisioning Decisions by Māori Caregivers to Ensure the Happiness and Health of Their Children

**DOI:** 10.3390/nu11050994

**Published:** 2019-04-30

**Authors:** Marewa Glover, Sally F. Wong, Rachael W. Taylor, José G. B. Derraik, Jacinta Fa’alili-Fidow, Susan M. Morton, Wayne S. Cutfield

**Affiliations:** 1School of Health Sciences, College of Health, Massey University, Auckland 0632, New Zealand; sallyfwong@gmail.com; 2Dunedin School of Medicine, University of Otago, Dunedin 9054, New Zealand; rachael.taylor@otago.ac.nz; 3Liggins Institute, University of Auckland, Auckland 1142, New Zealand; j.derraik@auckland.ac.nz (J.G.B.D.); w.cutfield@auckland.ac.nz (W.S.C.); 4Department of Women’s and Children’s Health, Uppsala University, 751 85 Uppsala, Sweden; 5School of Population Health, University of Auckland, Auckland 1142, New Zealand; jacinta@moanaresearch.co.nz (J.F.-F.); s.morton@auckland.ac.nz (S.M.M.); 6Centre for Longitudinal Research–He Ara ki Mua, University of Auckland, Auckland 1142, New Zealand

**Keywords:** Indigenous, nutrition, childhood obesity, social determinants of health, Māori health

## Abstract

Obesity in children is a global health concern. In New Zealand, one in three school entrant children are overweight or obese. Māori, the indigenous people, are disproportionately represented among the lowest economic group and have a disproportionately high incidence of obesity. This study explored Māori parents’ and caregivers’ views of the relative importance of weight to health, and the facilitators and barriers to a healthy weight in children aged 6 months to 5 years. Using a grounded qualitative method, in-depth information was collected in focus groups with mostly urban parents and other caregivers. A general inductive thematic analysis (content driven) was used. Insufficient money was an overriding food provisioning factor, but cost interacted with the lack of time, the number of people to feed, their appetites, and allergies. Other factors included ideologies about healthy food, cultural values relating to food selection, serving, and eating, nutrition literacy, availability of food, cooking skills, and lack of help. Childhood obesity was not a priority concern for participants, though they supported interventions providing education on how to grow vegetables, how to plan and cook cheaper meals. Holistic interventions to reduce the negative effects of the economic and social determinants on child health more broadly were recommended.

## 1. Introduction

Early prevention of malnutrition is important because the global increase in the prevalence of diet-related diseases appear to be beginning earlier in the life cycle. Preventative intervention early in a child’s development is critical, and has been argued to influence the biological systems [1] that appear to be as important in the pathogenesis of obesity by influencing how excess body fat is acquired [2].

Socio-demographic disparities in the prevalence of obesity in children are evident around the world. New Zealand is no exception, but has additional disparities by ethnicity. One in three children are overweight or obese by the time they start school (4–5 years of age), with ethnic disparities even at this young age [3]. Specifically, nearly twice as many Māori children—the indigenous people of New Zealand—are classified as obese (20%) compared to Europeans (12.7%) [3]. Māori are notably over-represented among the most economically deprived groups [4], which contributes to some of this difference. For example, 22.4% of children in quintile 5 (representing the 20% living in the most socioeconomically deprived areas) have obesity compared with just 9.7% in quintile 1 [3].

The obesity epidemic is systemic, but perhaps it may more correctly be referred to as ‘syndemic’, which is the co-occurrence of two or more epidemics [5]. Obesity is compounded for vulnerable groups by the effects of structural inequalities, such as income disparity [6] and institutional racism [1]. There have been increasing calls for approaches that are more holistic [7], collaborative, multi-sectorial, and include multiple stakeholders [8]. Design of salient and acceptable interventions, especially when working across different cultural groups [9,10], must include partnering with communities, understanding the specific challenges of their local food system [4], and co-creating solutions [11,12].

Parents play an important role as part of the wider food system. In order to design effective early interventions to support healthy child growth, it is useful to consider the perceptions of parents regarding their food provisioning practices, beliefs about food and healthy child weight, and the influence of the family environment [13,14]. Lack of time, the high cost of healthier foods [15], the relatively lesser cost of convenient fast foods [16,17], and a perceived lack of nutritional knowledge and skills [18] have been identified as major barriers to healthier food provisioning.

The aim of this study was to explore the views of Māori parents and other caregivers on the relative importance of weight to health, and the barriers to and facilitators of healthy weight in young children (6 months to 5 years of age). A focus of this paper is how decisions are made in the provision of healthy and unhealthy foods.

## 2. Materials and Methods

This study is part of a larger A Better Start National Science Challenge project (www.abetterstart.nz/the-science/healthy-weight/) that explores how New Zealand parents perceive children’s weight, and facilitators of healthy growth in young children. A qualitative study design using focus groups was chosen to explore views across a range of cultural groups. This paper presents the results from the Māori focus groups. A second paper will report cross-ethnic ranking of the influences on young children’s weight.

Focus groups are useful when seeking to understand the meanings different population groups apply to topics [19], and have been recommended for studying factors influencing food provisioning and for identifying culturally-specific etiologies of ideal body size [20]. When working with disempowered or minority groups, focus groups enable participants to support each other if they want to share less socially desirable opinions [19], thus helping to mitigate social desirability bias [20]. Focus groups also align with a common Māori data collection method: *hui*, which respects Māori values such as inclusivity, meaning that if more participants than intended attend they are welcomed [21].

A non-random purposive sampling framework [22] was used to recruit potential participants: principally Māori parents, but also grandparents and other caregivers of children aged 6 months to 5 years in the Auckland and Northland regions. Participants were recruited individually and through community groups serving lower socioeconomic areas.

The focus group facilitator was ethnically matched to enable cultural protocols to be observed and Māori language to be spoken if participants chose to do so. Focus groups began with a verbal explanation of the study and distribution of a participant information sheet, consent form, and short demographics questionnaire. Using a semi-structured focus group schedule, the group was prompted to discuss their views on healthy and unhealthy foods, how they define overweight, perceived reasons for excess weight among children, and what could be done to help parents provide healthier foods more often. The focus groups took 1½–2 h; they were audio recorded and subsequently transcribed. All participants were given a NZD 50 retail voucher as compensation for transport costs.

The style of facilitation encouraged a natural *korero* (talking) style of discussion, which allowed participants to range back and forth across the focus group topics. Thus, despite the study’s deductive beginnings (i.e., research design, questions, and analysis determined by research objectives) thematic analysis using a general inductive approach was used, allowing unexpected categories to be identified [23]. Transcripts were independently coded by two researchers (M.G. and S.F.W.). This involved reading each transcript to identify phrases or blocks of text with a distinct meaning and applying a label, such as ‘unhealthy weight’ or ‘healthy *kai* (food)’. The coding was compared and found to be largely consistent, with less than 3% discrepancies. These were discussed until consensus was reached. Category and theme labels were then plotted according to their relationship to each other to create a model, and enable the selection of a corresponding visual depiction of the model to assist in communicating the results [24].

Quotes in the text represent an excerpt from group discussion. Questions from facilitators are identified by a ‘Q’ and enclosed in square brackets.

This study was approved by the University of Auckland Human Participants Ethics Committee (#018082). All participants provided both verbal and written informed consent. Questionnaires and focus group transcriptions were kept anonymous prior to analyses. This study was performed in accordance with all appropriate institutional and international guidelines and regulations for medical research, in line with the principles of the Declaration of Helsinki.

## 3. Results

### 3.1. Participant Demographics

Five focus groups with a total of 37 participants were conducted (Table 1). Four focus groups were conducted in the Auckland region and one focus group was held in Northland (Whangarei).

Participants were all aged 20 and over, mainly women (89%), over half (54%) had a partner, and under half (43%) were engaged in full- or part-time employment (Table 2). Most participants self-identified as Māori or of dual Māori and other ethnicities (87%). Four women self-identified as Pacific in the East Auckland focus groups. Whilst the overall discussion from that focus group was included in the analysis, content specific to Pacific cultural practices and quotes from these four women have not been included in this paper.

### 3.2. Qualitative Results

A variety of factors affecting participants’ decisions about what to feed their children were identified. The participants discussed a few factors at a time, indicating that the factors were sometimes interdependent and interacted in a way that changed what food would or could be provided each day. The relationship between the factors and how they interact is depicted in Figure 1.

The model uses the metaphor of a *kete* (Māori woven flax basket) to convey the multiple interdependent factors influencing food provisioning decisions aimed at fostering happy children. The handle of the *kete* symbolizes the overarching restriction of how much money is available to buy food. The walls of the *kete* imply the interaction of several factors, such as how much time is available for food planning, preparation, cooking, serving, dining, and cleaning up. Time is moderated by how much stress the parents or caregivers are experiencing. How many people need to be fed must be factored in, as will their appetites, allergies, and taste preferences, which, in turn, influence what food will be acceptable. Different people consume different amounts of food, so the volume needed at each meal and for snacks needs to be calculated and provided for. Choice of foods is also influenced by people’s beliefs and values, which are moderated by culture. Different nutrition guidelines, conflicting information, and ideologies sway food choices. Furthermore, individuals and families vary in their nutrition literacy and skills at effectively organizing, budgeting, and cooking. Skill is moderated by the help available to complete food-provisioning tasks. The intended food choices represented by the whole content of the *kete* depend on the interactions among the above factors, which, in turn, are dependent on factors external to the family, that is, the type and cost of food items available in the community.

Each factor in this model is explained in the following sections.

### 3.3. Money Available to Buy Food

The amount of money available to buy food each day is an obvious influencing factor, but the focus groups revealed that it was not the dominant factor upon which all others relied. Income from work or a government benefit is paid as a lump sum, either weekly, fortnightly, or monthly. Participants reported that it was up to the parents to budget this amount to ensure they can feed their children every day until the next payday. A factor that influences daily decision-making and food availability is treats. Examples of treats include branded restaurant chain food, which cost a lot more than food from local takeaway shops. Another “treat” was buying better quality food in greater quantities in the days immediately after payday, even though these treats exhausted the budget for good balanced meals in the last few days before the next payday.

• “...you might start off after payday, the beginning of the week, good meals, and then the following week if you get paid fortnightly, like we get, my husband gets his salary paid monthly, so it’s like… Feast and famine… you have the whole month rather than, you know, just one week and the three weeks we’re eating crap.” (Whangarei)

All focus groups agreed that healthier foods, such as fresh fruit and vegetables, cost a lot more than unhealthy foods.
“The cost of healthy foods are just too expensive, so it’s just easier... to get takeaway. It’s cheaper to get a pie and fizzy drink, it’s $2 or $1, than make a salad. Yeah, than to make a nice lunch.” (East Auckland).“A lot of the cheapest food isn’t the most nutritional, so a lot of the kids binge on stuff that aren’t so good for them, like dollar pies or whatever the cheap stuff is at the bakery. Chips (fries). So, it’s not the healthy stuff… it’s expensive to eat healthy. Everyone knows that. If you wanna try and eat organic, or, the most expensive stuff in the grocery stores are the healthy food.” (North Shore 2).

### 3.4. Accommodating Multiple Different Appetites, Allergies, and Food Preferences

In addition to financial constraints, deciding what to have for dinner depended on how many people needed to be fed, the extent of their appetites, whether anyone had allergies, and what everyone’s food preferences were. Many parents talked about children refusing food, demanding certain foods, being “picky” and “fussy”, and “small” or “big” eaters.
“They’re going to get what they’re going to stretch their budget out to cover them and their kids. So, they can throw in a few goodies for the kids, but once that runs out they’ve got nothing else to eat but what the parents are eating, basically. And then that depends on that child’s, um, that’s child’s appetite. They might be a big eater, might be a little eater, and you don’t know. You’ve always got to balance that.” (South Auckland)“I’ve also got a thing about my sons, because our allergies, or intolerances, to dairy foods, I’ve gotta find alternatives to that as well, so. The time, and stuff, doing that is… They might go, we don’t like spaghetti Bolognese, or you know, we don’t like this and that, coz they’re the ones that initially waste all the food that you’re trying give them… and then if you don’t, if they don’t eat their dinner that’s when you find them in the pantry late at night time, when you go to bed, coz they’re hungry.” (North Shore 1)

Parents talked about how hard it was to choose healthy foods and to restrict consumption of ‘bad’ foods such as drinks, sweets, and chocolate, especially when other children were ‘spoilt’. The parents expressed frustration that other children were not similarly restricted, or that other people, particularly grandparents, fed children what the children wanted as well as special treats. A parent from Whangarei commented: “…grandparents, you know, rock up, take the kid to the shop and it comes back with a Lift Plus [fizzy drink] can.” The grandparents in each of the two focus groups where this was said confirmed that they did this, stating that it was part of their role and their right to treat their *mokopuna* (grandchildren).

It was especially difficult for parents who were exhausted from working or on a tight budget to deny their child sweets, ice cream, and cake. This was particularly the case at social events where other children were enjoying all the food on offer, leading their child to feel punished and the mother feeling mean and guilty.

• “My daughter’s only three, so from when she started eating solids I try to give her the right thing all the time. But her cousins are always eating chocolates and stuff and I didn’t want her to miss out.” (East Auckland)

### 3.5. Volume of Food Needed per Number of Meals and Snacks per Day

Decision-making about food provisioning is not done just once a day to decide what to feed the family for dinner. Parents have to work out how much food is needed for breakfast, snacks, lunch, dinner, and treats.

• “You just feed them because you have to, breakfast, lunch, and dinner, and snacks in between because you just constantly want to make sure that they’re not unhappy. So, you just feed them their fruits, their biscuits, whatever they’re meant to have.” (North Shore 1)

Only a few parents talked about planning ahead. One woman had to plan ahead because she lived out of the city, her husband got paid monthly, and that’s when they would do their one big monthly shopping. They were one of the few participants who said they grew their own vegetables.

### 3.6. Satisfying Food Preferences for Taste and Satiety

Taste and its importance to satiety must also be accounted for when deciding what to feed the family. This is particularly so if a family has budget constraints. They cannot afford to have people left unsatiated and wanting a second serving of food. In the following extract, a participant explains why a $1 pie is better than a jam sandwich when you are hungry.

• “You want to fill that gap there, you’re hungry and you want something that tastes good to fill that, that feeling, that feeling when you’re craving for certain foods. For example, you could go home and make you a jam sandwich, which is a lot healthier, um but you’ve got a dollar pie there. And the pie is going to fill that gap, the pie is going to, you know... Taste-wise, you know, yeah more satisfying.” (East Auckland).

### 3.7. Beliefs and Values Moderated by Culture

One man from Northland thought there were substantial cultural differences that determined what parents fed their children. He perceived that the relationship his *Pākehā* (New Zealand European) wife and her family had with food was “very minimal in terms of you know, food is there to sustain you, not to… celebrate as much with it... our [Māori] ideas of food were totally different.” He described a measure out of ten that his wife used for gauging when to stop eating, which others in the focus group could not fathom.

• “She’d eat up to like… a seven, like you feel like you’re a seven or something. And that’s when you stop eating. Had the pudding come out yet? …No! And her plate was still relatively full, you know, so she only ate until she was, you know… slightly uncomfortable. I wish I could do that. I’m going to eat til it’s all gone; I wish I had that willpower. Can’t waste that. Don’t waste that!” (Whangarei)

A value expressed across the focus groups was that it was bad to waste food. This was distinct from not wasting food due to a past experience of, or current fear of, food insecurity.

• “I wasn’t allowed to leave the table until everything was eaten, you know, and we had to eat our greens before we touched our meat. But we were full after the greens, but we f***ing worked for that meat, so we were gonna eat it whether we were full or not.” (Whangarei)

Some of the participants in the focus groups talked about the Māori protocol of *manaakitanga*, which is a cultural imperative and expectation to care for guests. Leaving guests wanting for food could result in *whakamā* (a sense of shame). This respect is reciprocal in that it is as important to graciously accept and enjoy what your hosts have gone to so much trouble to provide, as illustrated in the following quotes:“It’s definitely a cultural thing… with Māori people it’s always you fill up your guests.” (North Shore 1)“It is a culture thing as well… he’ll [her son] be 2 next Saturday… it’s a mind frame thing that you freak out. So, I’d easily spend a couple of hundred dollars on just the food for his party... Most parties I’ve been to they just have sausages, but they have like really minimal, like chips [crisps] and dips… My thing is making sure that everyone’s fed like really nice *kai* [food], but it’s always way too much... So, there does seem to be some kind of, um, overcompensation, or fear, fear that um, especially events… it is kind of like *manaakitanga* that you’re caring for your guests kind of thing. Whereas… most of the kid’s parties we’ve been to there’s nothing of the scale of, of the overcompensation that, that a lot of our *whānau* [families] events seem to, you know, have.” (North Shore 1)

Another strategy the *Pākehā* wife of the Northland man above had for ensuring their children ate healthy was to pack homemade foods, such as carrot and celery sticks, for her children to eat when they went to children’s parties. The following excerpt shows how this was perceived as “sad” by the Māori parents in the focus group:

• “Oh… didn’t you find that sad? Yeah, I did! I did. If my husband did that, I wouldn’t have a family left! [laughing] My family would be highly offended. [chorus] Yeah! Yeah, mine were! Brought their own *kai*!” (Whangarei)

Regarding culture, participants talked about the negative impact that colonization has had, and that globalization was having, in terms of the loss of knowledge and skill, such as growing their own food. However, they thought that many improvements in diet had occurred over the last 20 years, especially at the community level. They said that most *marae* (traditional meeting places) now served healthy meals and many *marae* had *māra kai* (communal food gardens) or arrangements with local food producers or landowners for access to be able to pick puha (native annual herb traditionally part of the staple diet) and watercress.

### 3.8. Nutrition Guidelines and Conflicting Information and Ideologies

Participants provided numerous examples of information that parents are exposed to, telling them what to eat and what not to eat. They discussed how confusing it was, and some spoke of the difficulties they experienced trying to implement various diets and dietary fads, such as switching to Himalayan salt, washing supermarket-bought food in cider vinegar to remove chemicals, eating a sustainable diet, and especially ‘clean eating’. Many parents were trialing changes to their family’s diet that were burdensome in terms of the higher cost and greater time required to obtain specialty foods and prepare them from scratch, despite the lack of evidence for claimed health benefits.
“The main fruit I usually get is apples. Whatever’s cheap at the time I’ll just get a bunch of that, which is apples and bananas. And then you know, you find out information that you shouldn’t, you know, you shouldn’t eat bananas at a certain time, and it’s just all this different information that comes out.” (East Auckland)“With this, like, transition into clean eating um and not having refined sugars and stuff, I’m not getting anything that’s processed. So, when I get home from supermarket shopping everyone’s just like: ‘Where’s the food?’ [laughing] …literally everything I have to bake. Like biscuits. So, there’s no packets of biscuits anymore.” [Q: And do you not put sugar in your baking?] “It’ll just be not refined, so either honey, or maple syrup, or dates. But it’s challenging coz if you don’t put the time and effort into baking as soon as you get your food back home, everyone’s just like, ‘What is there to eat?’…It’s hard.” (North Shore 1)

In addition to the clean eating diet requiring this mother to spend a lot of time baking, she was often unable to afford the foods she wanted to buy to adhere to the ideology:

• “I would like to be able to buy free range eggs and organic fruit and veg and poultry but there’s just no way… it’s too expensive. …On our weekly food money. There’s no way we can afford to eat the best produce… Like it’s just the money issue.” [Q: So you perceive the organic foods to be healthier?] “Um, totally! If I look at some of the people that are um, I don’t know, like breast cancer people that have just used nutrition to be healthier or combatted it through nutrition rather than chemo, they eat super healthy and it’s organic.” (North Shore 1)

Some parents talked about feeling bad about not being able to feed their children “properly”. Food insecurity occasionally led parents to encourage their children to “eat it all” because, as they explained, they weren’t sure when they would be able to serve, for example, chicken again, “or when your next meal is going to be”. Across the groups, parents described that they often felt guilty about this form of food insecurity. An example was the mother who sometimes didn’t have enough money for meat and fresh vegetables, saying that “ideally I don’t wanna give her a whole plate of pasta.”

• “When I relief teach, I get, you know, $200 a day, which is boom! $200 on the food for the next fortnight. But when I’m relief teaching I don’t have time to make the healthy foods, so I do, do the, the packet popcorns and the packet chips.” (Whangarei)

### 3.9. Time Available Is Moderated by Stress

Time available for planning meals, obtaining the food, preparing it, cooking it, serving the meal, and cleaning up afterwards varied depending on a parent’s life circumstances, the number of children in their care, and their work situation (e.g., full-time, part-time, shift work). Lack of time was not a simple barrier to healthier food provisioning, as competing demands for that time was also a consideration. For example, working all day and wanting to spend time with the children, but needing to prepare dinner instead. In this context, stress was frequently put forward as an explanation for not providing healthy food.

• “Just getting home from work and preparing dinner… And your kids aren’t feeling ya that day, and it’s just rush, rush, and you just wanna get them into bed.” (North Shore 1)

Some participants believed that a number of parents who did not feed their children healthy foods had “addictions” or they had their priorities wrong. For instance, parents who were quick to leave their children with family members so they could spend time going to festivals, and parents who were more interested in partying, drinking alcohol, and using drugs.

• “Drugs is a big thing… the cost of one night of ‘woohoo!’ is the cost of, you could feed your family for a week and half. And gas [petrol].” (Whangarei)

### 3.10. Capability to Plan Meals, Cook Food, and Nutrition Literacy Is Moderated by Help

The groups debated what could be done to assist parents to improve the nutritional value of the food they provide for their children. The important skills and capabilities discussed included meal planning, shopping on a budget, how to cook, what to cook, recipes, how to grow food, and how to read nutrition content labels on food packages.

• “So, cooking classes, *wānanga* [educational workshop or course], um, about what to eat, how to cook, what’s healthy? … And just even like how to cook for the budget as well. Yeah. Coz that’s a big thing eh, having the money to have that, you know. A nutritious meal for less than 15 dollars… Less than 15 min as well, you know. Yeah. And a full meal, yeah.” (Whangarei)

Some Auckland participants had benefited from such programs, while others believed that these programs did not exist in their area (e.g., Whangarei). Participants commented that to be attractive, education programs needed to be interactive, free, and provide childcare, or, even better, involve the children in the learning. However, requiring parents to go somewhere would have limited reach, due to several barriers to participation. These included parents being time poor, financially stretched, and lacking in childcare or support to attend. The effectiveness of these programs was also questioned. As one parent said, not everyone has time to read labels, they need to be able to trust that if food is on the shelf, then it is good enough for people to buy.
“I did parenting programs, I did cooking… like at the Anglican Church where they do like cheap budgeting meal ideas… I think there’s programs out there, but it’s more the parents having to go and source that as opposed to people just saying, go to this.” (Whangarei)“And food ideas, meal ideas. Yeah, I was going to say, quick and easy ideas… With pictures… we got a menu book, and it had like the yummiest like, something like satay chicken salad, it was yummy, it was yummy as, and it was quick and easy to make… healthy recipes… Gardening. How to garden. How to grow your own veges… And how much exercise for them to burn off stuff. Coz you know they’re like, if you eat one biscuit it’s so many hours. But in children, because their metabolisms are so fast. We think a 20 min play at the park, right that’s enough! …how long should they, should they be having their exercise… for winter and summer, coz like it’s different?” (North Shore 1).

However, educational programs and interventions were seen by some participants as “reactive rather than proactive”; an “ambulance at the bottom of the cliff”. They believed that what was needed was to address the economic determinants of food insecurity. As many participants said “it all comes down to cost.”

Consistent with this idea was joining a co-operative or facilitating families to set up food co-ops to bring down the cost of food. This could be achieved by buying in bulk from farmers or distributors, and/or sharing surplus fruit and vegetables. Access to cheap fruit and vegetable stalls every day after work highlighted the need for access to healthier food choices, which are as convenient as the fast food shops. South Auckland participants felt especially “saturated” with the latter. One focus group said there were some social media groups that enabled access to “cheap organic meat, cheap organic whatever.”

Not having help to prepare meals compounded the size of the task. If the job of preparing healthy meals began to feel too burdensome, then the easier, and generally less healthy, option became more attractive: “it’s so much easier getting fish and chips. Yeah. And you just sit there and everyone helps themselves.” (East Auckland)

### 3.11. Access to Food, Type of Food Available, and Cost of Food

Access to affordable food is a significant factor in food provisioning decisions by Māori parents. A key point raised by one participant was the perceived greater cost of healthier foods compared to unhealthy foods.

• “There’s a difference between supermarkets and vege shops you know. Veges are more cheaper than the supermarket itself. Yeah. Where are you going to choose, you know, where you’re going to go shopping. And it’s gotta be within your budget, so you know you can afford to get all that stuff.” (South Auckland)

A cultural change in New Zealand was that previous generations used to put excess homegrown fruit or vegetables out for others to take, and children used to ‘raid’ neighbourhood fruit trees. “Now they’re like “what are you doing?” The free fruit available for kids in certain supermarkets was appreciated.

Several groups talked favourably about different schemes that provide financial assistance to buy food, or provide access to discounted foods. It was also noted that people donated food that went into food parcels. However, some of the participants who had been recipients of food parcels said that sometimes the food was “not really healthy” and it was often past its best-before date.

• “That food you can get from church parcels or from [charity], it’s outdated, its use-by dates are already up… but what do you do? You know, they’re a [charity], they need food to feed all the homeless.” (South Auckland)

Food grants were also accessed by some participants, but occasionally these were used to pay other bills. One woman said she traded her food grant card for cash at a local convenience store so she could pay her electricity bill. She expressed outrage that the store owner charged her $20 for doing this, but she felt she had no choice, and the rest of the group sympathized and agreed with her.

When asked what could be done to help parents with healthier food provisioning, the South Auckland group suggested: “It’d be good if they could subsidize healthy foods.”

Growing vegetables as a way of increasing access and reducing cost was rarely mentioned in the focus groups. A few individuals had gardens, but most did not. There was interest in having help to set up vegetable gardens at home, but the barriers to growing their own vegetables were lack of knowledge about how to grow food, insecurity of housing tenure, and the cost and time involved in setting up and maintaining a garden. Furthermore, some participants thought that Housing NZ (a government agency and the country’s largest provider of rental accommodation) did not allow tenants to have vegetable plots. Communal gardens were not seen as a practical solution in urban South Auckland. One group believed that the Council had tried planting fruit trees and vegetable patches in public areas previously, but these had been removed due to the high maintenance cost of cleaning up fallen fruit. They also thought that such schemes would be abused by shopkeepers who would pick the fruit or vegetables to sell in their shops.

The previous sections identified and described interacting factors that influence food-provisioning decisions by Māori parents. A principal aim of this study was to determine the views of Māori parents and caregivers on the relative importance of weight to health. This topic and a related issue, the role of exercise in child health, are explored in the following two sections.

### 3.12. Happiness, Not Weight, Should Be the Focus

Having satiated, and thus happy, children was what was most important to parents and caregivers in these focus groups. Weight was not perceived to be an appropriate focus for research, especially when some children were perceived to be under-nourished.

Many parents did not have bathroom scales in their house, and if they did, they tended to use them for weighing themselves, but not their children. Children were only weighed to determine a medicine dosage or to ascertain if a child was underweight.

Participants were not knowledgeable about obesity—they thought it only applied to people who were seriously overweight—and nor was there widespread acceptance that obesity in children aged under 5 years was a problem. While there was acceptance that some children might be considered ‘big’, the parents believed this was not problematic. Their view was that once the children started school, their activity levels would increase so much that they would burn off excess fat. These views are illustrated by this comment: “When they’re that young they kinda just grow into their body as they get older.” (North Shore 2).

The only concern about childhood obesity was the risk of giving children an emotionally and psychologically damaging complex about their size, so that they might develop eating disorders when older. They worried that stigmatizing children based on their size or weight created a permissive environment for bullying of bigger children and discrimination against them. Parents and caregivers in this study believed every child was different, every child was special, and it was harmful to compare children to each other.

One brief story revealed the deep divide between Māori traditional beliefs about children and child-rearing, and the contemporary pressures that are undermining Māori traditions. A participant who worked at a *Kōhanga Reo* (a Māori preschool language and education center) spoke about the compassion they felt for a child who had been put on a diet. They disagreed with what they perceived as punitive, restrictive, and isolating treatment of the child, which led them to replace the child’s carrot stick lunch with what the other children were eating.

• “We had one girl in the *Kōhanga* I was working in, and because her parents were overweight, obese. Overweight they were. They put their child on a diet and that was torture for us to see the rest of the kids eating yummy *kai*, and she’s got like carrot sticks and, you know, all this other healthy stuff… So, we’d just like put her lunch away and give her a cake! [laughing] Because that is torture, it really is. You’re punishing the child because of your actions, you know? Because you’re overweight, you’re scared your child’s gonna get overweight, so you put them on a diet. Like, it’s a child. It’s torture for them.” (South Auckland)

The most important focus in terms of child wellbeing was that children were not sick, and that they were developing physically, cognitively, and emotionally as well as could be expected. A healthy child was happy, active, and, as described by one South Auckland grandmother, “nice and open and they’re not scared... And they speak their mind.” The parents advised not intervening when there was not a problem.

### 3.13. Barriers to Sufficient Physical Activity

Depending on the context, parents balance the often incompatible demands and goals identified above in the process of deciding what to feed their children each day. A non-nutritional goal that parents felt compelled to achieve for their children was to ensure that they had enough exercise.

Some parents felt they needed more information on how much exercise was enough relative to the child’s food consumption. Barriers to healthy food provisioning—time, money, preferences, and information—also undermine parents’ decisions to exercise their children. They reported that there seemed to be few organized community events for physical activity, sports, or services that catered for toddlers and young children. Public swimming pools and private indoor child parks were available, but for regular activities, the latter were reportedly cost-prohibitive. As one parent summed it up: “It’s just too expensive to do anything.” (Whangarei)

One focus group knew of a community trust that provided “an inside winter preschool play thing” four mornings a week in different suburbs each day, but “getting there” was a problem. If exercise programs did exist, they would need to be offered after normal business hours to cater for working parents. During the winter, there was an additional barrier, as parents tended to keep their children away from these public places and events because they believed this increased their child’s risk of catching colds and flu from others:

• “…the spread of germs. You go to the pools in the weekends and the kids in there and all their snot in the pool… Avoiding getting sick.” (North Shore 1)

Related to levels of physical activity, parents expressed a realist attitude towards the reliance on TV and tablets as “baby sitters”. This was especially the case for mothers with two or more young children, some of whom also looked after other people’s children during the day.

## 4. Discussion

Consistent with calls for a more holistic understanding of the systemic and cultural determinants of health and risks to health, such as childhood obesity, this study found that the food provisioning choices of Māori parents and caregivers were influenced by a wide range of interacting and differentially weighed factors. Cost, tiredness, stress, lack of help, time required for food preparation, varied food preferences, and demands or likelihood of children refusing food, these and other factors all impacted on parental food provisioning decisions.

Previous studies [15,16,25] have cited the cost of healthy foods as a major barrier to healthier diets. However, our study shows that the price of healthy foods cannot be looked at in isolation. Cost is relative to the many other factors presented in our model, such as access and preparation time. Further, cost is often researched in a reductionist way, for instance, when it is defined in terms of the affordability of food items relative to other food items, such as healthy versus unhealthy [16]. The amount of time necessary for travelling to a store and shopping for food is often overlooked [15], though one study attempted to account for the additional cost incurred in food preparation time [16]. Our participants identified time as a major barrier to provision of healthy food. Food preparation time was weighed relative to their desire to spend quality time with their children, as well as their general sense of having insufficient time for most things in their daily lives.

The interaction of food prices and the value of time involved in planning, sourcing, and preparing meals have been associated with the rise in obesity. Chou et al. [26] identified the increasing scarcity and increasing value of household and non-market time as a determinant of an increased demand for fast food and consumption of food away from home. The societal shift to both parents working has raised family income and reduced time for preparing meals at home. This has increased demand for pre-prepared, takeaway, and restaurant meals. In addition, technological innovations and the realization of economies of scale have enabled the fast food industry to lower the price of fast foods, while also conveniently situating outlets in areas of greatest demand. Our research supports other literature that suggests food, at least healthy food, is highly replaceable. That is, the amount and quality of food will be compromised when other needs must be met first. Examples include paying the power bill, buying medicines or medical treatment [27], or when grandparents restrict their own food consumption to be able to feed their grandchildren [17].

A strength of our study is that it provides qualitative detail that supports holistic models such as the Foresight Obesity System Map [28] that demand recognition of the many systemic, structural, and contextual factors driving health outcomes and the effectiveness of interventions. Interventions to prevent or reduce childhood obesity need to account for their impact on each of the influencing factors to avoid unintended negative effects [29]. For example, an intervention that focuses on just one factor, such as reducing fast food outlet density in lower socioeconomic suburbs [30], could inadvertently increase the cost of food provisioning because parents incur increased travel costs to buy food.

A persistent challenge in finding appropriate ways to reduce obesity is the frequent mismatch between a medical focus on preventing excess weight gain and the parents’ priorities for the child’s wellbeing [13,31,32]. Hassink (2017) concluded there is a pressing need to improve clinicians’ understanding of the modifiable and protective factors [33]. The factors identified in this research could be developed into a measure to assist health professionals working with families to pinpoint the specific factors having the greatest negative influence on healthier food provisioning. Any intervention with parents and caregivers needs to recognize that they may have different and potentially incompatible goals, such as wanting to prepare healthy food for their children, but also wanting to spend more happy time with their children after work [34]. Potential solutions could be identified and discussed with the goal being to redirect the food budget towards healthier foods, whilst reducing, or at least not increasing, the time involved in food provisioning. For example, the mother who was trying to implement the clean eating diet would have benefited from expert advice that informed her that it was okay to feed her children produce that is not organic; thus saving her some money and guilt. She also was expending precious time baking from scratch to avoid refined sugar, but the sucrose substitutes were as high in energy as sugar. She had the will and commitment to feed her children the healthiest food possible, but she lacked accurate information on how to do this within her budget. Measuring the influence of these factors across a population of families in a defined area could also highlight community specific problems, such as a suburb lacking access to physical activity opportunities for toddlers or healthier fast food options. Community responses and strategies for mitigating the negative effects of structural determinants of health can then be facilitated.

Māori parents are not unaffected by trends in dietary ideologies, such as the idea that people should be eating sustainable diets. The in-depth discussions in this study showed that food provisioning decisions were made more difficult by conflicting and sometimes false information. For example, some parents thought healthy food was only ‘fresh food’, which sometimes cost more than what they could afford. Frozen, dried, or canned foods were being overlooked as other healthy alternatives, with the result that fast and cheap foods (i.e., high in calories, but low in nutrients) were being purchased. Trustworthy information on what food is healthier and on how to quickly produce tasty, filling, ‘bang-for-buck’ dishes for a large family would be very useful for these parents.

Notably, there are unintended consequences of inducing high levels of parental guilt and loss of efficacy among caregivers, especially lower socioeconomic parents, about their food provisioning decisions. For instance, Fielding-Singh [6] found that when caregivers eventually receive their income, they try to make up for the hard days of doing without by over-compensating. For example, they may treat their children to expensive fast foods or chocolate. Similar results are reported in this study, in food provisioning decisions immediately after a payday.

It is important to note that food is a medium for the expression of culture [6]. Healthier food interventions should respect and support cultural beliefs, values, and protocols around food [9]. In the New Zealand context, this includes, for example, *manaakitanga* and *whakanoa* (the sharing of drink and food after a formal activity to ‘make common’—to be on a level footing with each other). This is practiced both in traditional Māori meetings and at home, particularly when a stranger visits for the first time. Discouraging the eating and celebration of culturally significant foods or dishes, or trying to change the dishes themselves, risks perpetuating a power–culture dynamic that sees *Pākehā* cultural systems transmitted through public health policy and programming [35]. Our research suggests that it is everyday meals that should be the focus of healthier food interventions, not events and festivals that celebrate and preserve culture. Interventions should support Māori tribal and community self-management to counteract the dependency created by colonization and the associated ‘cultural imperialism’, which have prevented Māori capacity to establish culturally relevant *tikanga* (codes) for living that protect against modern threats to health [36]. Interventions delivered locally into the home or local community, also minimize transferring the cost of interventions to the families themselves, which needs to be avoided because any drain on the family budget reduces the capacity to provide healthy food.

Differences between past generations raised on farms and contemporary urbanized Māori were suggested as one reason why younger urbanized Māori parents were less likely to know how to grow food. This could be mitigated if the centrality of the extended family, specifically the influence and role of grandparents, was harnessed by interventions. Tapera et al. [17] found grandparents have gardening and healthy food provisioning knowledge and the willingness, if enabled, to moderate a number of the negative influences identified in our focus groups.

Our study identified a number of economic and social determinants of healthier food provisioning that could be targeted. Though out of the range of typical obesity prevention interventions, the complex system of determinants, competing behavioural change demands, and failure of reductionist approaches [1,35] focused on single points of intervention in isolation, warrant the need to work inter-sectorally to employ ‘proxy’ interventions that can deliver downstream beneficial effects on health [8]. Given the finding in this study that early childhood obesity is not a priority, interventions addressing determinants identified by Māori are more pressing. Interventions that deliver improvements in diet and increased physical activity should be supported, and our participants identified a number of promising ideas (e.g., accessible education programs, food co-operatives, and reasonably priced public pools).

There was widespread interest in growing vegetables at home. Perceived barriers were insecure tenure of tenancy, lack of know-how, and a belief that most landlords prohibit gardens on their properties. A potential public health intervention would be for Housing NZ to support tenants to grow their own fruit and vegetables. For example, Housing NZ properties could be planted with a range of low maintenance fruit trees.

Food banks and community food pantries are benefiting from the willingness of many people and stores to donate food. It would be beneficial to provide guidelines to aid agencies, churches, and supermarkets on how to improve the nutrient quality of donated foods.

For low socioeconomic families, the cost of electricity is another barrier to the provision of healthier foods. Primarily, the price of electricity draws funds away from the food budget. The electricity bill gets paid first as otherwise it will be disconnected, causing additional disconnection and reconnection fees to the family’s already outstanding debt. Furthermore, many cooking methods require the use of electrical appliances, which for some families may be an additional barrier to cooking. As previously mentioned, one participant reported having to sell her food vouchers so that she could pay her electricity bill. Of note, when parents and caregivers limit the use of electricity at home there are much wider implications for public health, with heaters not being used, the hot water cylinder being turned off, not vacuuming the house, or not using the washing machine or drier as often.

### Limitations & Strengths

A limitation of our study was that fathers and grandfathers were under-represented. In addition, the majority of our participants lived in Auckland (New Zealand’s largest city), and while one focus group was conducted in Northland, most of those participants also lived in urban areas. Thus, the results may not be generalizable to other Māori families living in other regions, and particularly to those living in small towns and rural areas. Though recruitment focused on people living in lower socioeconomic areas, socioeconomic status of participants was not measured. Over 40% of participants were employed and over 50% had some post-school qualification. The results are therefore unlikely to represent only Māori living in the more deprived areas.

Much of the writing about healthy diets and healthy lifestyles assumes universal agreement of what these are. However, international nutrition guidelines vary and many dietary recommendations and health claims remain contested [37]. These guidelines may reflect the dominance of a monocultural aspiration. In New Zealand, for instance, a lifestyle modelled by a low body mass index (BMI) is consistent with the views of middle-upper class New Zealand Europeans who maintain relatively higher levels of physical activity and are generally better able to afford premium food produce. The use of the universal BMI across ethnicities is also not without opposition [38]. Lastly, healthy and unhealthy foods were not defined by the facilitators, thus the use of these terms by participants does not necessarily mirror nutritional guideline definitions.

Our study had a number of strengths. Focus group discussions enabled the collection of rich in-depth information. Whilst the overall number of participants was relatively small, the involvement of grandparents and other caregivers added to the diversity of views presented. It was a strength that the groups were facilitated by a Māori researcher, allowing culturally specific terms to be used and attendance to cultural protocols for meetings. From a research perspective, a naturalistic method enabled clarification of statements to gain a deeper understanding.

Lastly, we showed that there are many interacting and often conflicting factors affecting the food provisioning decisions made on a daily basis by Māori parents. The complexity of how these facilitators and barriers to healthier eating interact requires thinking and analysis that go beyond the normal range of initiatives, which aim, for example, to address the global increase in rates of childhood obesity. It is essential that any healthier food interventions adopt a holistic approach that is culturally appropriate, giving adequate consideration to the complex reality experienced by families.

## Figures and Tables

**Figure 1 nutrients-11-00994-f001:**
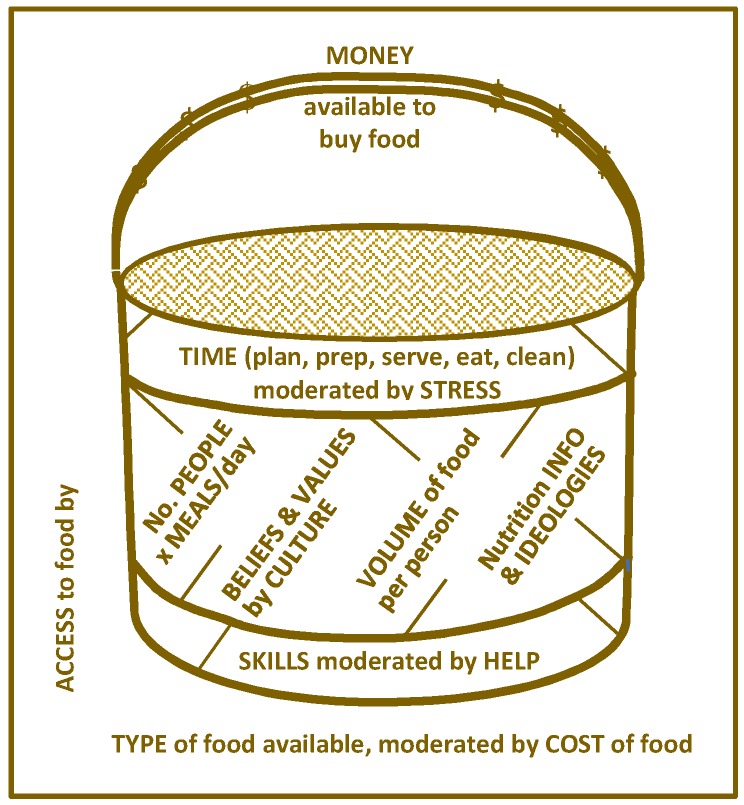
*Kete* (basket) of factors influencing food provisioning aimed towards having happy children.

**Table 1 nutrients-11-00994-t001:** The composition of Māori focus groups.

Group	Participants
East Auckland	2 Māori mothers, 4 Pasifika mothers
South Auckland	5 Māori mothers, 1 Māori grandmother
Whangarei (Northland)	13 Māori parents, 2 Māori aunties, 1 Māori grandmother
North Shore 1 (Auckland)	7 Māori parents
North Shore 2 (Auckland)	5 Māori parents

**Table 2 nutrients-11-00994-t002:** Demographic profile of focus group participants (n = 37). Where appropriate, data are n (%), mean (range), or median (quartile 1, quartile 3).

**Gender**	Female	33 (89.2%)
	Male	4 (10.8%)
Age (years)	20–29.9	14 (37.8%)
	30–39.9	11 (29.7%)
	40–49.9	8 (21.6%)
	≥50	4 (10.8%)
Ethnicity	Māori	32 (86.5%)
	Samoan	2 (5.4%)
	Tongan	2 (5.4%)
	Cook Island Māori	1 (2.7%)
Country of birth	New Zealand	32 (86.5%)
	Overseas	5 (13.5%)
	If born overseas, time living in New Zealand (years)	20 (9, 28)
Education	No school qualification	5 (13.5%)
	High-school qualification	9 (24.3%)
	Post-school qualification (trade, diploma, or certificate)	12 (32.4%)
	University degree	9 (24.3%)
Marital status	Single/never married	12 (32.4%)
	Married/de facto/civil union	20 (54.1%)
	Widowed	1 (2.7%)
	Separated or divorced	4 (10.8%)
Employment status	Full-time or part-time employed	16 (43.2%)
	Student	7 (18.9%)
	Homemaker	7 (18.9%)
	Not currently employed	6 (16.2%)
Partner’s employment status	No partner	5 (13.5%)
	Full-time or part-time employed	15 (40.5%)
	Not currently employed	14 (37.8%)
	Homemaker	3 (8.1%)
	Both parents not currently employed	3 (8.1%)
Household	Number of adults	2.4 (1–5)
	Number of children	2.9 (0–10)

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
