# Peer review of "The Complexity of Food Provisioning Decisions by Māori Caregivers to Ensure the Happiness and Health of Their Children"

_nutrients, 2019, doi:10.3390/nu11050994_

Round 1

Reviewer 1 Report

This paper is well written and represents a gap in the use of qualitative methods to understand food preferences and cultural norms around eating and obesity in the Maori community. 

Intro

line 42- explain highest quantile of neighborhood deprivation (e.g. socioeconomic measure?)

line 46- expand a bit more on syndemics and why it is a useful frame for obesity

Methods

when was data collected?

lines 94-100 move to discussion

Was a semi-structured protocol used by the facilitator? This is unclear.

I have some concern over including Pacifika mothers in this analysis due to relatively small sample size and focus on Maori caregivers. Pacifika mothers may have very different experiences related to food compared to Maori groups. If you do keep them in, please include some background about how these food behaviors, preferences, cultural norms, etc might be comparable. 

Table 2 contents- left align instead of centered

line 190- make sure not to over state grandparents responses since it looks like there were only two...include this sample size in this comment

Discussion

Do you have any data related to socioeconomic status or neighborhood deprivation level of these participants? Since you talk about these disparities in the intro, it would be helpful to know if the sample is representative of families from high deprivation areas. If this sample was higher income, please state and modify introduction to better align.

Did you see any differences in responses related to access, cost, time, etc. based on families living in the city versus northland? 

The last section of the discussion could be improved by restructuring as 'implications for practice' and including all recommendations based on the findings in this section and emphasize key takeaways. Alternatively, you could better align the discussion with the theoretical formula so it is easier to follow.

Author Response

Please see the MS attached.

Reviewer 2 Report

This study explores food provision choices of Maori parents and caregivers using a qualitative approach: focus groups, grounded theory and thematic analysis. The study is part of a wider research programme exploring how parents in New Zealand perceive chlid weight and the facilitators of healthy growth in children.

The manuscript presents a research design appropriate to the research question. I believe the research will be of interest to the readers of Nutrients by highlighting the complex interactions between individual and systemic drivers of food choice, as well as the importance of social and cultural values that may influence dietary aspirations and traditions. I also think it is wise to remind readers to be cautious with absolute BMI ratings in young children and to not underestimate perceptions of what it means to be healthy.

I support the publication of this manuscript in Nutrients upon some minor revisions. In the main these relate to a fuller introduction to set the scene, a little further interpretation of the data, and changes to the layout of the results to clarify the findings. I hope the below are of use to the authors and wish them every success with the revisions.

[2] Title: Is your main take home message the complexity of food provisioning decisions, or the notion that weight is not necessarily a good determinant of health and happiness, or do you feel both are necessary? Clarifying this might allow you to home in on the main thread of the paper and could be how you focus the presentation of your results.

[22] Abstract [and throughout]: Clearly state which research questions led the current study [these can be within the context of a wider study] and be consistent throughout e.g., whether the study was aiming to look at "relative importance of weight to health" "facilitators and barriers to a healthy weight" "how decisions are made in provision of healhty and unhealthy foods" "how NZ parents perceive children's weight, facilitiators of healthy growth in children" "views of healhty and unhealhty foods" "define overweight" "reasons for excess weight in children" "help parents provide healthier food more often". Could these be grouped or presented so that it is clear what the overarching aims were and the main concepts focused on in this study? If the aims are i) "relative importance of weight to health" ii) "facilitators and barriers to a healthy weight" then it might aid the reader to also present the results in this way.

[25] If you have the word count it might be useful, for Nutrients readers unfamiliar with qualitative research, to put content driven in brackets after "inductive thematic analysis".

Introduction: Set the problem of childhood obesity and current concerns. Is this a problem acute at that age or is it more the tracking of obesity into adulthood and associated health outcomes in adulthood? i.e., if childhood obesity disappeared what would be the health impacts? Are there differences in health outcomes between Maori and NZ Euro populations in adulthood? Is obesity an effective marker in young children and what are the current controversies about using weight per se or BMI as a risk measure in young children and in children of 2019 who might be from different reference populations than the orginal used to create the BMI risk profiles. You could also introduce the [69] "culturally specific etiologies of ideal body size" point raised in the methods section. This might all lead nicely to why you believe it is necessary to explore the importance of weight to health.

[41] The economic part of your introduction is important to your research question on the "barriers and facilitators of healthy weight": you could easily frame your whole results around the notion that food choice and health outcomes have a complex relationship with SES and many other interacting variables. Tighten language "which likely accounts" and use citations to clarify point.

[52] Consider rephrasing "must focus on parents". Your earlier text relating to the system and structural inequalities is great. It might be undermined somewhat by putting the focus only on individual parents and their behaviours to effect change. I would rephrase to emphasise parents play an important role as part of the wider system etc. and this is why it is useful to know their perceptions.

[62] Materials & methods: I would be interested to know if you expected to publish a comparitive piece: Maori focus groups vs, NZ Euro focus groups?

[82] Do you need to explain why the results have not been presented by focus group topic - that these led the discussion and the thematic analysis looked across topics to identify themes?

[94] Rephrase "scientifically there is no shared understanding" paragraph. I think the point of this paragraph is valid in terms of the general public perception of health being nuemerous and at times tacit/difficult to articulate. No shared understanding would leave you open to criticism when there is consensus on many areas, if only certain nutrients are required in differing amounts to sustain growth and bodily/brain function, variety of fruits and vegetables; energy intake being sufficient but not consistently surpassing energy expenditure etc.

Results: I would suggest adding a sub-heading to separate the descriptives from the qualitative results. It might be helpful to, at the start of the results section, sum up the results in one paragraph - an orientating paragraph so that the reader is pre-warned as to what to expect: the layout of the results etc.

[118] Consider using "a healthy child is a happy child" as a traditional theme and use the information presented in other areas to explore your research questions around the notion of weight and healthiness and values that are important to parents in terms of child development. 

[124] I praise the authors for their adventure in creating such an equation but I would omit it from this paper. I do not know how useful it is above and beyond the presentation of the qualitative data. At present it is unclear how and why certain concepts have/have not been included or categorised together nor why they have been arranged in such a way. Where is taste, is it part of NA? How does this equation relate to other theories of behaviour/attitudes? if the authors are interested in this area then I would suggest an additional piece of research that involves a group modelling exercise or something similar to devise such as formula/representation of food chocie complexity.

Throughout results: Please look again at the themes you have presented [and the quotes] and consider if there is overlap or if these are the best representation of your results. There are lots of interesting findings, I think just a little more time to interprete the data and arrange the results in fewer clearer categories [which could have sub-headings also] would make the results section easier to follow. Do not feel that you have describe every item in your dataset, nor the most common, just the most meaningful in the context of your research.

[275] I suggest not to over interpret results and make sure you stay as close to the data as possible.

[409] This could be interepreted as parents were more concerned about calories than the quality of the diet - is this the case?

[430] Discussion: Check this does not make the participant identifiable.

[505-511] The resource is unclear, I suggest rephrase this paragraph to incorporate the shared responsibiity of both individual and structural determinants of health - to align with the introduction.

[526] Should this sustainability section be in the results rather that the discussion?

The discussion is focused on parental interventions, if this is the purpose of the paper then I would suggest more clearly orientating the whole paper around the "help parents provide healthier food more often" research question?

Author Response

Please see the MS attached.
